# Using Colour Images for Online Yeast Growth Estimation

**DOI:** 10.3390/s19040894

**Published:** 2019-02-21

**Authors:** Elias August, Besmira Sabani, Nurdzane Memeti

**Affiliations:** Institute of Chemistry and Biotechnology, Zurich University of Applied Sciences, 8820 Wädenswil, Switzerland; sabanbes@students.zhaw.ch (B.S.); memetnur@students.zhaw.ch (N.M.)

**Keywords:** software sensor, non-invasive online measurements, pattern recognition, computer vision, optical density measurements, automatisation

## Abstract

Automatisation and digitalisation of laboratory processes require adequate online measurement techniques. In this paper, we present affordable and simple means for non-invasive measurement of biomass concentrations during cultivation in shake flasks. Specifically, we investigate the following research questions. Can images of shake flasks and their content acquired with smartphone cameras be used to estimate biomass concentrations? Can machine vision be used to robustly determine the region of interest in the images such that the process can be automated? To answer these questions, 18 experiments were performed and more than 340 measurements taken. The relevant region in the images was selected automatically using K-means clustering. Statistical analysis shows high fidelity of the resulting model predictions of optical density values that were based on the information embedded in colour changes of the automatically selected region in the images.

## 1. Introduction

Today’s trend in many laboratories of the biochemical and biotechnology industries as well as in academia is towards automatisation and digitalisation. This trend is part of the so called fourth industrial revolution, where readiness and transparency of data play an important role. Crucial for basic research as well as low cost production is cultivation in shake flasks, for which, however, means for automatic data retrieval, particularly, online measurement techniques, are mostly lacking. On the one hand, such measurements are necessary for monitoring and control of system variables such as pH, biomass, and temperature, and, ultimately, for process automation and optimisation. On the other hand, they shall be such that their impact on culture conditions and growth is as minimal as possible [1,2,3].

At the end of the day, what industry is interested in during biomass cultivation is greatest possible yield. Optical density (OD) of the medium is known to be a good proxy for biomass and, thus, is often used to estimate growth [4], as direct determination, for example, through dry mass samples is much more elaborate, time consuming and introduces considerable lag, which makes this technique unsuitable for monitoring, control and automatisation in general. OD is obtained by measuring the medium’s absorbance of light at a specific wavelength, which, by the Beer–Lambert law, is proportional to the concentration of a corresponding component. Until recently, OD was not measured online during cultivation in shake flaks. In 2014, Ude and colleagues presented in [2] a device for online measurements of pH, pO${}_{2}$ and OD in shake flasks. It consists of a sophisticated prototype platform with an OD biomass sensor based on backward light scattering. They successfully calibrated measurements against cell dry mass and, now, market it through PreSens Precision Sensing GmbH, Regensburg, Germany.

Since then, further work on online measurement of OD has been conducted and includes, for example, portable devices for OD measurement in micro-fluidic chips [5] and for low light detection of bioluminescence [6], where both applications are smartphone-based. Indeed, smartphones as affordable means have been recently repeatedly employed for colour-based bio-sensing [7], as they are more accessible and cheaper than analytical laboratory devices [8], particularly, for reading of pH indicator paper [3,9,10]. Other examples of their employment in the field are detection of environmental contamination in water samples [11] and of allergen contamination in foods [12]. The latter lies within the trend towards point-of-care diagnostics using a portable, inexpensive, and user-friendly platform for the detection of bio-targets, the so-called ‘Lab on a Chip’ [13,14,15]. Notwithstanding the progress made with respect to smartphone-based colorimetric tests, most applications require external housing units or various accessories, which increases device complexity and makes the use more difficult.

In this work, we are interested in affordable and user-friendly means for non-invasive measurement of biomass concentrations during cultivation in shake flasks, where the produced data can be stored in electronic format, particularly, in so-called electronic laboratory notebooks, which offer the possibility to store all data in one place and clearly annotate them. Moreover, the system shall be easily extendable at a low cost for measuring other system variables such as pH (for example, by detecting change of indicator paper colour) and pO${}_{2}$ (for example, by counting air bubbles). We envision the use of such a system in the educational sector and for prototyping, for allowing sterile means of measurement, being low cost, and enabling process automatisation and optimisation. Importantly, it shall not be confined to cultivation in shake flasks and generally applicable to monitoring and control in single-use technologies, which are becoming increasingly popular in terms of economy, convenience, and quality [16]. More specifically, this paper presents an online OD software sensor, where we investigate the following research questions:Can images of shake flasks and their biomass content acquired with smartphone cameras be used to estimate biomass, just as anecdotical evidence that experienced technicians can do so through visual inspection only suggests?Can machine vision be used to robustly determine the region of interest, the content of the shake flask, in the images such that the approach suggested for estimating biomass can be automated?

To answer those questions, first, we repeatedly cultivate yeast in shake flasks and, every 30 min, take samples for measuring OD and acquire images of flasks. Then, we proceed as follows:We determine a regression model for predicting OD through the colour components of the region of interest in the images and assess its prediction performance.We apply K-means clustering to segment each image into distinct sections at the start of each experiment and manually mark the one that corresponds to the region of interest. Thereafter, segments of the subsequent image are compared to it automatically and the most similar one chosen for use in predicting OD and for comparison with segments of the next image and so on.

## 2. Materials and Methods

In this section, we first present the yeast strain used and how it has been cultivated in shake flasks in Section 2.1. Section 2.2 is about colorimetric methods. In Section 2.2.1, we briefly describe how we obtain OD measurements to estimate biomass concentrations. Next, we detail the image acquisition procedure in Section 2.2.2 and introduce the concept of a colour space, into which images are mapped, in Section 2.2.3. To elucidate distinctive features in the images, we apply the K-means clustering technique in the colour space, which is explained in Section 2.3. We seek to isolate a specific region in the image, which corresponds to the medium in the flask. We use the values of colour components in this region to predict OD values through a linear regression model as described in Section 2.4. Finally, we evaluate our approach using the statistical methods presented in Section 2.5. Throughout the paper, all algorithms and analyses are implemented using MATLAB${}^{®}$ Release 2018b, The MathWorks, Inc., Natick, Massachusetts, USA.

### 2.1. Yeast Cultivation

A 1 L shake flask with 250 mL of Yeast Extract Peptone Dextrose (YEPD) medium consisting of 10 g/L yeast extract (Carl Roth, Karlsruhe, Germany; 2363.4), 5 g/L peptone (Merck, Darmstadt, Germany 95039-5kg-F), and 10 g/L glucose (Brenntag Schweizerhall AG, Basel, Switzerland; 81585-330) is autoclaved at 120 ${}^{\circ }$C for 20 min. The shake flask of the pre-culture is inoculated with thawed and mixed cryocultures (2 mL cryovials with 40% glycerol at −80 ${}^{\circ }$C) of *Saccharomyces cerevisiae*. To prevent foaming, three to four drops of a sterile 20% PolyPropylene Glycol (PPG) solution are added under sterile conditions. The yeast cells are cultivated in the incubator at 30 ${}^{\circ }$C and 160 rpm for 11 h and then stored at 4 ${}^{\circ }$C for inoculation in experiments to follow, where the same conditions and media are used. Thus, with inocula obtained from the pre-culture, the above procedure is repeated for all subsequent 18 cultivations. Table 1 summarises how the experiments differ with respect to inocula volume-size and the location, where images of the flask were acquired (see Section 2.2.2). Note that, for the reminder of the paper, since in most experiments the exponential growth phase ends after about 9.5 h, to avoid fitting noise when obtaining the regression model (Section 2.4), we omit the last 1.5 h in the investigation.

### 2.2. Colorimetry

Generally speaking, colorimetry’s goal is to specify the concentration of coloured compounds in the medium. One approach is doing so through OD measurements (Section 2.2.1). Here, we aim at obtaining the same information through colour changes in images of the content of the shake flasks (Section 2.2.2 and Section 2.2.3). Figure 1 depicts the colour change of the medium during cultivation.

#### 2.2.1. Optical Density

OD is measured with a CECIL 1011 spectrophotometer (Cecil Instruments Limited, Cambridge, United Kingdom). The wavelength, at which absorbance is measured, is set to 600 nm and the YEPD medium at timepoint t=0 h is used as reference, that is, the spectrophotometer is then reset. Every half hour, a 2 mL sample is taken under sterile conditions; the macro cuvette that contains it is placed in the photometer, and OD is determined and recorded. When OD readings exceed 0.7, the sample is diluted with water (distilled water with 9 g/L NaCl).

#### 2.2.2. Image Acquisition

Images are acquired at two locations in the laboratory with different backgrounds, one which is white and provides a rather bad contrast (Figure 2a) and another which is blue and provides a rather good contrast (Figure 2b), as this colour is not expected to be much expressed elsewhere in the images. In addition, the light conditions are different, location 1 (tabletop) is rather unshielded and, thus, arguably more sensible to changes in light than location 2 that constitutes the floor under the bench. We use two different smartphones, iPhone 7 and iPhone 6S by Apple Inc. for location 1 and location 2, respectively. Both are equipped with a 12 MP camera with High Dynamic Range (HDR), where the former one provides images of size 1200×1600 pixels, while the latter provides ones of size 768×1024 pixels, which we consider to be qualitatively equivalent for the purpose of the paper. After incubation, four images are acquired in quick order every 30 min. Notably, we take care for the focus to be right, and images are always acquired from nearly the same distance, and the smartphones are held steadily at nearly the same angle, for which we appreciate the help of a PopGrip from PopSockets (Boulder, Colorado, USA) (Figure 2c). Nevertheless, images still quite differ as shown in Figure 3, which highlights the challenges for an image-segmentation procedure and for a method that uses changes in a medium colour to predict changes in OD.

#### 2.2.3. Colour Space

A colour space, of which different ones exist, is used to numerically describe colours. We convert images into the CIE 1976 (L*a*b*) (CIELAB) colour space, which approximates human vision and perception of lightness. Its development has been driven by scientific theory on how the brain translates colour stimuli. Thus, this colour space seems best suitable for a method that is derived from visual inspection by humans. In the CIELAB colour space, colour is defined by three dimensions. It has dimension *L* for lightness and dimensions *a* and *b* describe the green-red and blue-yellow colour components, respectively [17]. In MATLAB${}^{®}$, image information is transformed into the RGB colour space using imread and into the CIELAB colour space employing rgb2lab. At each measurement timepoint, for all four images, we compute the mean value of each colour space dimension of the entire image as reference as well as of the region of interest and then, for all six variables, we take the mean of all four images for use in making predictions of corresponding OD values. An exemplary result is shown in Table 2, where for the *i*-th measurement, we denote the OD value by $y_{i}$, the colour components of the reference by $L_{ref_{i}}$, $a_{ref_{i}}$, and $b_{ref_{i}}$ and those of the region of interest by $L_{i}$, $a_{i}$, and $b_{i}$.

### 2.3. K-Means Clustering

The objective of this method is to group together similar numerical data, where similarity is defined through the Euclidean distance. Particularly, K-means clustering aims at partitioning *N* observations into *K* clusters, where cluster members have in common that their distance to cluster centre $\mu_{k}$, the mean of all their positions, is relatively close, k=1,2,…,K. Formally, our goal is find those values of $r_{nk}$ and $\mu_{k}$ that minimise the following objective function

(1)$$J=\sum_{n=1}^{N}\sum_{k=1}^{K}r_{nk}{∥x_{n}-\mu_{k}∥}^{2}.$$

Here, $r_{nk}$ indicates whether data point $x_{n}$ belongs to cluster *k*; that is, $r_{nk}=1$ if data point $x_{n}$ is assigned to cluster *k* and $r_{nj}=0$ if j≠k. To invoke the K-means clustering algorithm, we use kmeans in MATLAB${}^{®}$ with all options set to default. For more details on the algorithm, see reference [18]. In the following, we describe the algorithmic procedure we employ to determine the region (cluster) of interest in each image, where observations correspond to pixels that are described in the CIELAB colour space through a three-dimensional vector; that is, cluster centre $c^{\left(i\right)}$ has entries $c_{1}^{\left(i\right)}=L_{i}$, $c_{2}^{\left(i\right)}=a_{i}$, and $c_{3}^{\left(i\right)}=b_{i}$.

Importantly, Algorithm 1 below expects *K*, the number of clusters, as input. A method that serves us well to determine the number of clusters is the elbow method [19]. The number of clusters is chosen such that adding an additional one does not increase (much) the information gained or, more formally, such that the ratio of distances between cluster centres to within-cluster distances of all clusters ceases to change (much). Initially, one observes a steep curve when this ratio is plotted but soon the curve flattens out rather abruptly. This creates an “elbow” in the graph, after which not much change occurs. Note that, for speed, we applied the method to the first few images from both locations only, observed “elbows” in the graphs for K≤7, and then fixed *K*. We chose K=9, a slightly higher number for “safety” instead of repeating the procedure for each and every image. Indeed, visual inspection showed instances of unsatisfactory segmentation for six or seven clusters.

**Algorithm 1:** Determining the region of interest in images.**Input**: *K*.Set i=0.**Repeat** for all experiments.
**Increment *i***: Set i=i+1.Segment first image, $im_{1,i}$, acquired at timepoint t=0.0 h into *K* clusters using K-means clustering.User chooses cluster of interest. Its corresponding cluster centre is denoted by $\tilde{c}$.Set $\bar{c}=\tilde{c}$.**Repeat** for t=0.0 h, 0.5 h, 1.0 h, *…* until t=9.5 h.
-**Reset *j***: Set j=0.-**If t>0.0 then** set i=i+1.-**Repeat** at each measurement timepoint for all four images.
○**Increment *j***: Set j=j+1.○Segment image $im_{j,i}$ into *K* clusters using K-means clustering.  ○Determine ${\hat{c}}^{\left(j\right)}$, the cluster whose centre is closest to $\bar{c}$ by means of the Euclidean distance.-**Update**$\bar{c}$ and $c^{\left(i\right)}$: Set $\bar{c}=\frac{1}{4}\sum_{j=1}^{4}{\hat{c}}^{\left(j\right)}$ and $c^{\left(i\right)}=\bar{c}$.-**Store $c^{\left(i\right)}$**.


#### Example

Figure 4 is exemplary of applying K-means clustering to an image, where the first or upper-left panel corresponds to the region of interest, the content of the shake flask.

### 2.4. Linear Regression Models

Let us denote the set of all measurement by M whose elements are given by tuples of the form $m_{i}=\left\{c^{\left(i\right)},y_{i}\right\}$. To predict the OD value of the *i*-th measurement, which is $y_{i}$, from the values of the CIELAB colour components, we use multi-variate linear regression models of the following form:(2)$${\hat{y}}_{i}=\beta_{0}+\beta_{1}x_{1}^{\left(i\right)}+\beta_{2}x_{2}^{\left(i\right)}+\ldots +\beta_{9}x_{9}^{\left(i\right)}.$$

Here, ${\hat{y}}_{i}$ corresponds to model prediction of $f(y_{i})$, the dependent variable, where function f(·) transforms values of $y_{i}$ if necessary for obtaining a linear relationship between dependent and independent variables (Section 2.5.2). The independent variables are given by $x_{j}^{\left(i\right)}$, j=1,2,…,9. Variables $x_{1}^{\left(i\right)}$, $x_{2}^{\left(i\right)}$, and $x_{3}^{\left(i\right)}$ are the possibly transformed values of dimensions *L*, *a*, and *b*, respectively, of cluster centre $c^{\left(i\right)}$; in other words, $x_{1}^{\left(i\right)}=g_{1}\left(c_{1}^{\left(i\right)}\right)$, $x_{2}^{\left(i\right)}=g_{2}\left(c_{2}^{\left(i\right)}\right)$, and $x_{3}^{\left(i\right)}=g_{3}\left(c_{3}^{\left(i\right)}\right)$, where the transformations are denoted by functions $g_{1}\left(\cdot \right)$, $g_{2}\left(\cdot \right)$, and $g_{3}\left(\cdot \right)$. Next, $x_{4}^{\left(i\right)}=g_{1}\left(c_{1}^{\left(20\cdot \ell +1\right)}\right)$, $x_{5}^{\left(i\right)}=g_{2}\left(c_{2}^{\left(20\cdot \ell +1\right)}\right)$, and $x_{6}^{\left(i\right)}=g_{3}\left(c_{3}^{\left(20\cdot \ell +1\right)}\right)$, where $\ell =\left\lfloor \frac{i}{20}\right\rfloor$ and ⌊·⌋ is the flooring function; that is, $c^{\left(20\cdot \ell +1\right)}$ is the cluster centre associated with the first measurement of the corresponding (ℓ+1)-th experiment, as 20 measurements are performed per experiment. The latter is to account for baseline colour of the medium (that is, without or only little biomass) that might differ between experiments. Finally, variables $x_{7}^{\left(i\right)}=g_{1}\left(L_{ref_{i}}\right)$, $x_{8}^{\left(i\right)}=g_{2}\left(a_{ref_{i}}\right)$, and $x_{9}^{\left(i\right)}=g_{3}\left(b_{ref_{i}}\right)$. They serve as reference values to account for changing light conditions during and between experiments.

### 2.5. Model Performance Assessment

#### 2.5.1. R-Squared Measure

A common measure for the goodness of the fit is known as R-squared. It accounts for the amount of variation in the target variable explained by the model and is calculated as follows, where $y_{i}$ is the data to be modelled (here, OD values), $\bar{f}$ is the mean of all $f(y_{i})$, function f(·) is as in Equation (Equation 2), and ${\hat{y}}_{i}$ are model prediction of $f(y_{i})$:(3)$$R^{2}\equiv 1-\frac{\sum_{i}{\left(f\left(y_{i}\right)-{\hat{y}}_{i}\right)}^{2}}{\sum_{i}{\left(f\left(y_{i}\right)-\bar{f}\right)}^{2}}.$$
Values of R-squared close to 1 are considered good fits, that is, most variability in the data is captured by model predictions. Note that the differences given by $f\left(y_{i}\right)-{\hat{y}}_{i}$ are known as *residuals*.

#### 2.5.2. Regression Diagnostics: Test for Linearity and Homoscedasticity

A plot of dependent versus independent variables is a useful visual test for the relationship between the two. If nonlinearity is evident, one should consider applying a nonlinear transformation to the dependent variables and possibly also to the independent variables. After the transformation, in the plot of residuals versus predicted values, the points should be symmetrically distributed around a horizontal line and with a roughly constant variance.

#### 2.5.3. Regression Diagnostics: Test for Independence

Prediction errors or residuals are expected to be random and uncorrelated such that correlation between an array obtained by ‘stacking’ them and another obtained by ‘stacking’ their time-shifted version, called *autocorrelation*, is low; otherwise, the model can be probably improved [20].

#### 2.5.4. Regression Diagnostics: Test for Normality

If normality is violated, then the computed confidence intervals for the predictions are questionable. We use the one-sample Kolmogorov–Smirnov test to test whether data is normally distributed.

#### 2.5.5. Cross-Validation

We use cross-validated $R^{2}$-statistics to assess prediction performance of our model. Cross-validation helps to prevent overfitting by dividing the data into training and testing datasets. Particularly, for all *ℓ*, ℓ=1,2,…,18, we train model *ℓ* using observation data from all experiments but from experiment *ℓ*. Then, using measurements obtained for experiment *ℓ*, we predict OD values from colour components through Equation (Equation 2). The prediction performance is assessed by means of the R-squared measure. To estimate autocorrelation of residuals, we employ leave-one-out cross-validation on randomly ordered data, which means that we repeatedly use measurements obtained at all timepoints but one to estimate it through Equation (Equation 2).

## 3. Results and Discussion

### 3.1. K-Means Clustering in the CIELAB Colour Space

Figure 5 is exemplary of applying Algorithm 1, where we show only nine images. In general, visual inspections of all results confirmed the correctness of the algorithm.

Note that, in general, there isn’t a guarantee that Algorithm 1 will always correctly segment all images or identify the region of interest, particularly, when the variability between images is greater, for instance, because they are acquired onsite in the shaker, possibly while still in movement. Nevertheless, a semi-automatised process is possible through manually either readjusting the choice of the image segment to focus on or, if the clustering algorithm fails, cropping the image such that only the region of interest remains.

### 3.2. Test for Linearity and Homoscedasticity

Figure 6 shows the relationship between OD and colour components of all experiments. The three panels to the left show data as is, and the right ones show data, where all values have been log-transformed, which is $f\left(\cdot \right)=g_{1}\left(\cdot \right)=g_{2}\left(\cdot \right)=g_{3}\left(\cdot \right)=\log\left(\cdot \right)$ in Equation (Equation 2), where we omitted the first measurement of each experiment, as for timepoint t=0 h, the OD value is zero and, thus, cannot be log-transformed. Clearly, the three panels on the righthand side rather depict a linear relationship between dependent and independent variables. Thus, we consider a linear relationship between log-transformed values of OD and log-transformed values of the colour space for the regression model. For the reminder of the paper and all experiments, we consider the measurement at timepoint t = 0.5 h to be the first measurement.

### 3.3. Model for Predicting Future OD

Assuming a linear relationship between log-transformed values of OD and log-transformed values of the colour space that is expressed through a model of the form given by Equation (Equation 2), we obtain predictions for OD-values using the cross-validation scheme described in Section 2.5.5. Visual inspection of Figure 7, the plot of residuals versus predicted values, seems to reveal slight violation of linearity and homoscedasticity. However, the Kolmogorov–Smirnov test did not reject the assumption that residuals are normally distributed.

When employing leave-one-out cross validation as explained in Section 2.5.5, autocorrelation of residuals is within the confidence bounds of ±0.108, the mean of the residuals is roughly 0.01, and $R^{2}=0.81$. Figure 8 presents measured OD values and their predictions using the scheme developed in this paper, where we leave one experiment out when obtaining model parameters. Notably, again, the mean of the residuals is roughly 0.01, and $R^{2}=0.81$; that is, 81% of variations in the dependent variable are explained through independent variables. Additionally, $R^{2}=0.80$ for images from location 1 only and $R^{2}=0.89$ for those from location 2 only. Clearly, this result favours striving for a setting such as the one in location 2, if permitted by the laboratory environment.

## 4. Conclusions

In this paper, we presented a methodology for predicting OD, and ultimately biomass, during yeast cultivation in shake flasks by using information embedded in colour images of the flask’s content. To assess it, yeast was cultivated in shake flasks 18 times. Every time, after incubation, for 9.5 h and every 30 min, OD was measured and images of the flask were acquired. Importantly, we employed a rigorous statistical analysis that indicates high fidelity of the OD predictions based on colour information from the images. Additionally, the clustering technique presented in Section 3.1 allowed for automatically selecting the relevant region in the image, the content of the shake flask, from which to elucidate the colour information required by the model to make predictions. Thus, we provide a proof of concept for the usability of colour images for the estimation of biomass and for the automatisation of the method, for example, through the use of cameras fixed in place inside or outside the incubator. Clearly, the approach is restricted to neither yeast nor shake flasks and can be applied during cultivation of other organisms such as *Escherichia coli* in single-use bioreactor bags, for example. Moreover, in the future, we will also investigate whether the approach applies well to biomass, which is still in movement in the shaker, and to closed loop control of growth. In summary, the technique developed in this paper is simple to use, non-invasive, low-cost, can be readily automated, and provides data that is ready and transparent. Notably, the measurement system presented consists only of a smartphone and a few lines of MATLAB${}^{®}$ code, where, instead, free software such as R can also be used. As with all techniques that measure biomass indirectly, it only requires proper calibration against mass samples.

## Figures and Tables

**Figure 1 sensors-19-00894-f001:**
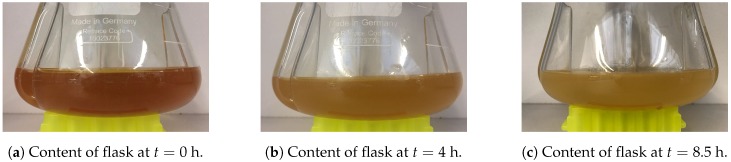
Colour change of medium during cultivation of yeast in shake flasks.

**Figure 2 sensors-19-00894-f002:**
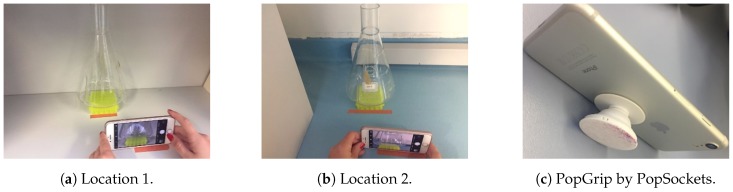
Locations, where images of shake flasks were acquired, and the PopGrip used.

**Figure 3 sensors-19-00894-f003:**
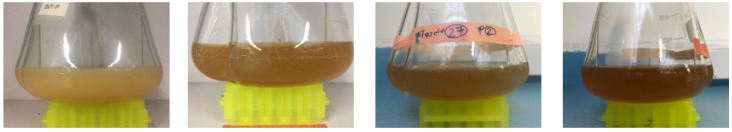
Exemplary images to highlight the variability among those acquired.

**Figure 4 sensors-19-00894-f004:**
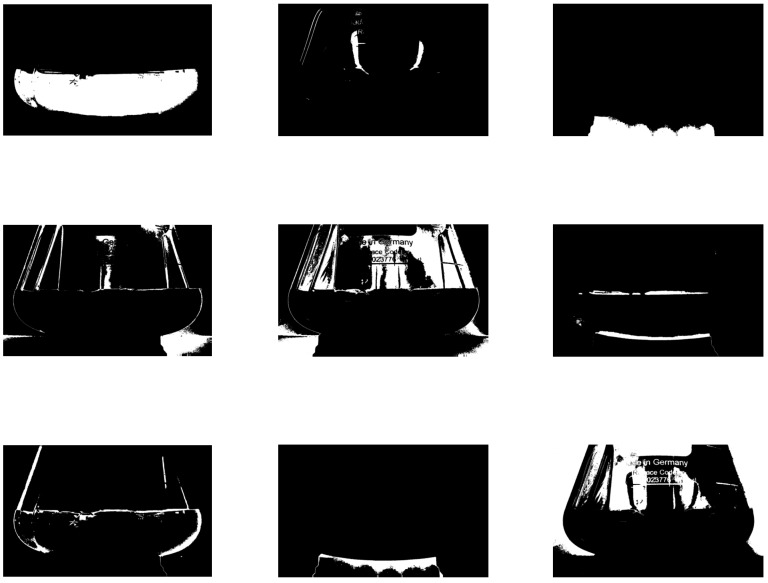
Result of applying K-means clustering to an image.

**Figure 5 sensors-19-00894-f005:**
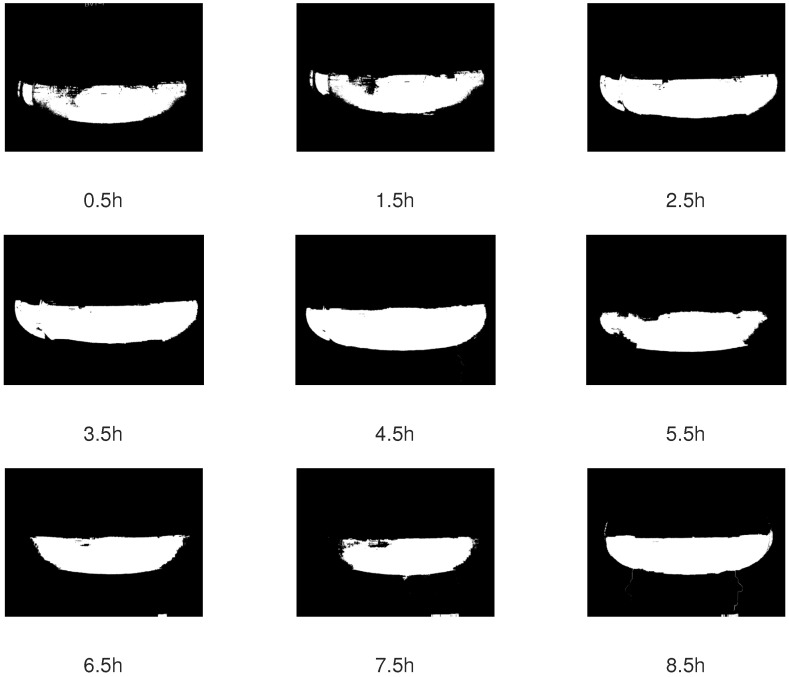
Result of applying Algorithm 1.

**Figure 6 sensors-19-00894-f006:**
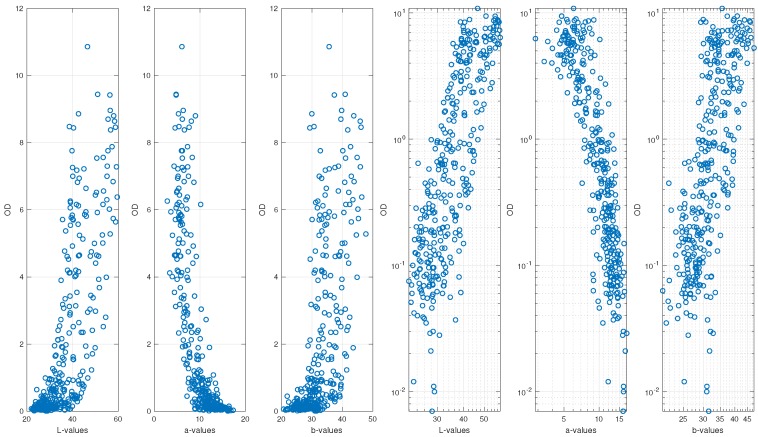
Relationship between optical density (OD) and colour components of all experiments. The three panels to the right depict values in the log-space.

**Figure 7 sensors-19-00894-f007:**
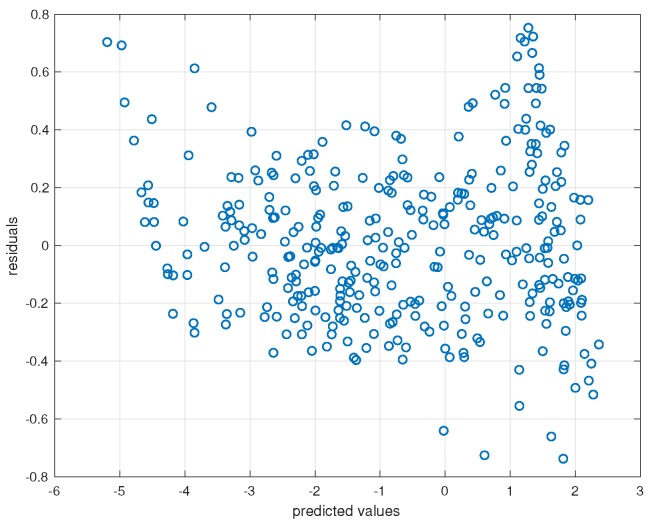
Residuals versus model predictions.

**Figure 8 sensors-19-00894-f008:**
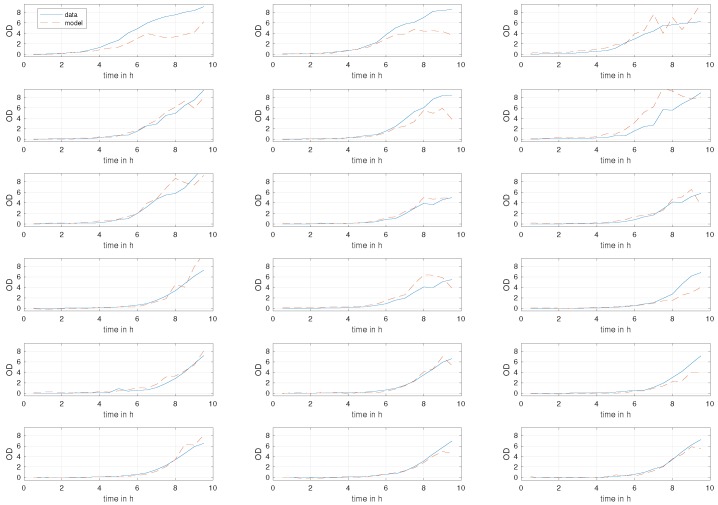
Observations and model predictions versus time.

**Table 1 sensors-19-00894-t001:** Summary of experiments: Inoculum size in ml and experiment location are shown.

**Experiment #**	1	2	3	4	5	6	7	8	9	10	11	12	13	14	15	16	17	18
**Size in ml**	15	15	15	10	10	10	10	10	10	10	10	10	10	10	10	10	10	10
**Location**	1	1	2	1	2	1	2	1	2	1	2	1	2	1	2	1	2	2

**Table 2 sensors-19-00894-t002:** Measurements 241 to 260 of CIELAB colour components and optical density (OD) values.

Time in h	$L_{\mathrm{ref}}$	$a_{\mathrm{ref}}$	$b_{\mathrm{ref}}$	*L*	*a*	*b*	OD
0.0	57.18	2.81	12.78	28.25	14.18	28.82	0.00
0.5	54.04	4.39	14.09	26.83	14.46	27.55	0.01
1.0	53.00	3.71	15.33	27.37	14.15	26.75	0.02
1.5	54.35	5.14	16.27	28.52	15.30	29.13	0.03
2.0	52.92	4.49	15.90	25.93	14.44	28.42	0.04
2.5	53.38	3.71	16.07	24.81	14.90	28.06	0.05
3.0	55.08	2.49	15.50	27.21	13.60	28.92	0.06
3.5	54.51	3.63	15.52	26.50	14.37	27.56	0.08
4.0	56.08	2.61	15.78	28.97	13.12	28.35	0.13
4.5	56.96	3.47	14.76	28.14	13.28	28.61	0.15
5.0	56.49	3.19	15.48	29.88	12.16	28.90	0.89
5.5	55.86	1.71	13.02	28.98	10.47	29.39	0.40
6.0	58.23	1.38	15.12	32.38	9.00	30.09	0.59
6.5	57.56	1.97	16.99	32.73	10.50	32.23	0.64
7.0	56.13	2.36	16.32	34.49	9.28	33.16	1.10
7.5	56.23	0.24	13.41	35.22	6.19	32.16	1.88
8.0	58.41	0.77	17.27	38.79	7.18	35.62	2.83
8.5	56.47	2.21	16.30	38.83	6.00	33.98	4.09
9.0	56.32	1.74	15.46	40.24	5.50	32.35	5.66
9.5	58.49	1.04	16.27	44.93	5.16	35.66	7.18

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
