# Peer review of "Using Colour Images for Online Yeast Growth Estimation"

_sensors, 2019, doi:10.3390/s19040894_

Round 1

Reviewer 1 Report

1) In section 2.2.2 you should specify the lighting conditions and background used, as well as temperature

2) In section 2.2.3 you state that the color space "mimics eye vision or human perception", but the CIE La*b* color space actually provides a color space that is perceptually uniform relative to the human visual system - this is a similar idea, but somewhat different

3) For this type of application you need some color normalization and the use of the reference values for the La*b* conversion will assist with this, but for completeness you really need to have information about the lighting and camera sensors. Or, it is possible, that you have experimentally determined that this is not necessary (?).

4) In section 2.3.1 (Algorithm 1) it unclear why the user selects the cluster of interest when the goal is to automate cluster selection

5) In section 3.1, paragraph two, it is unclear why the algorithm does not correctly segment the image. This is a fairly simple segmentation problem. I easily found a method using CVIPtools. It appears that correct object (segment) could be automatically selected by connected component with largest area.

Author Response

Dear Reviewer, our response is enclosed below. Kind Regards. Elias, Besmira, and Nurdzane

Reviewer 2 Report

General remarks:

1. Why the number of clusters in KM is equal to 9? For a smaller number of clusters, the program would run faster.
2. What about about details of initialization in K-means ?
3. The authors have completely forgotten to consider the effect of intensity and color of lighting on the measurement results.

Editorial remarks:

1. In the colour science, the term "colour space" is used, not "colour-space".
   On the other hand, the correct spelling is "colorimetry" and not "colourimetry".
   Similarly, "colourimetric" should be written as "colorimetric".  

2. Line9: The reader of the article may not know what "OD" means.

3. Line64: I propose "colour components" instead of "colour-space values".

4. Line73: "image acquisition" instead of "image taking".

5. Line101: "wavelength" instead of "wave length".

6. Line128: "Euclidean" instead of "euclidean".

7. Line164: Use italics for L, a, b.

Author Response

(The authors gave the same response as above.)

Reviewer 3 Report

The proposed work falls in the domain of automatisation and digitisation of laboratory processes, by proposing the measurement of biomass concentrations based on a images acquired by a mobile phone.

How is the number (9) of clusters in k-means clustering determined? Is this a heuristic finding appropriate only for the particular experiment or is it theoretically justified and applicable to other experiments as well?

In the presentation of colorimetric methods there is no mention of illumination requirements. As color is co-determined by illumination of a scene, there should be at least a requirement for constant illumination across different measurements (images). What other requirements do the experiments pose on illumination, i.e. regarding its spectrum and intensity?

The term online is mentioned only in the title and introduction. There is no mention of online measurement in the rest of the manuscript. It appears that images are first stored on the mobile device and then processed offline on a computer running Matlab.

The segmentation of the fluid in figure 4 is rather inaccurate, which is probably the result of segmentation through k-means which does not account for spatial relationships between segmented pixels. Nevertheless, the presented results could suffice as initialization for a more sophisticated algorithm i.e., [C. Rother, V. Kolmogorov, and A. Blake, GrabCut: Interactive foreground extraction using iterated graph cuts, ACM Trans. Graph., vol. 23, pp. 309–314, 2004.].

The experimental findings are not compared against ground truth (i.e. measurements obtained through conventional or gold-standard methods). There is no comparison or comparative discussion with other approaches in the literature.

A more typical term for image taking is image acquisition.

Author Response

(The authors gave the same response as above.)

Round 2

Reviewer 3 Report

I find that the comments of the reviewers have been addressed.